# Effects of Co Addition on the Properties and Microstructure of Cu-Ni-Si-P-Mg Alloys

**DOI:** 10.3390/ma14020368

**Published:** 2021-01-13

**Authors:** Jianyi Cheng, Fangxin Yu, Fu Huang

**Affiliations:** School of Materials Science and Engineering, Nanchang University, Nanchang 330031, China; chengjianyi@ncu.edu.cn (J.C.); 401329118024@email.ncu.edu.cn (F.H.)

**Keywords:** Cu-Ni-Co-Si-P-Mg alloy, aging, hardness, electrical conductivity, microstructure

## Abstract

Cu-Ni-Si alloys are widely used in electrical and electronic industry owing to excellent electrical conductivity and strength. A suitable addition of Co in the Cu-Ni-Si alloys can improve its strength and deteriorate its electrical conductivity. In this work, Cu-Ni-Co-Si-P-Mg alloys with different Co content are employed to investigate the effects of Co on the properties and microstructure. The results showed that Co addition lead to the formation of (Ni, Co)_2_Si precipitates. (Ni, Co)_2_Si precipitate is harder to coarsen than δ-Ni2Si during aging. The larger the Co content in the alloys is, the smaller the precipitates formed is. There exists a threshold content of Co to divide the studied alloys into two groups. One group of theses alloys with <1 wt.% Co or Co/Ni ratio <0.56 has the same aging behavior as the Cu-Ni-Si-P-Mg alloy. On the contrary, the time to reach the peak hardness of aging for another group can be obviously delayed and its electrical conductivity decreases slightly with the increase of Co content. It can be attributed to the lower diffusion rate of Co than that of Ni in the Cu matrix. Meanwhile, the Co addition can inhibit the formation of P-enriched Ni-P phase in Co-containing alloys during aging. The as-quenched Cu-1.6Ni-1.2Co-0.65Si-0.1P-0.05Mg alloy can reach 257 HV and 38.7%IACS after aging at 500 °C for 3 h, respectively.

## 1. Introduction

Cu-Ni-Si alloys are widely used for semiconductor lead frames in the integrated circuit packages and connectors in electrical and electronic industries because of their low cost, high strength, excellent bending workability and stress relaxation resistance properties, especially, good electrical and thermal conductivity [1,2,3,4,5]. The good electrical conductivity and strength is attributed to the precipitation of nanosized stoichiometric δ-Ni_2_Si intermetallic compounds [6,7,8,9]. In recent years, a further enhancement of the strength and a superior bending workability are required for the copper alloys used in electronic devices and in automobile components due to the high integration, high precision and reduction in size and thickness of electronic parts [10,11]. Therefore, some attempts have been made to improve the performance of Cu-Ni-Si alloys by increasing their Ni and Si contents and/or adding alloying elements such as Cr [3], Ti [12,13,14], V [15], Al [16], P [17], and Zr [18]. Besides, the addition of Mg in Cu–Ni–Si-based alloy can improve the mechanical property of alloy due to the Mg-atom-drag effect on dislocation motion [7,19]. However, these methods generally reduce the electrical conductivity of the alloys somewhat.

Recently, Izawa and Monzen et al. demonstrated that the addition of Co element to Cu-Ni-Si alloy can decrease the inter-precipitate spacing and increase dislocation density [11]. Zhao et al. found the multi-aging process and addition of Co can effectively improve the electrical conductivity of the Cu-1.01%Ni-0.28%Si-0.14%Co-0.45%Cr alloy [20]. Li prepared Cu-Ni-Co-Si and this alloy exhibited excellent comprehensive properties, including a hardness of 318HV, an electrical conductivity of 37.1%IACS, a tensile strength of 1003 MPa, a yield strength of 929 MPa, and an elongation of 4.5%, after the introduction of a multistage thermo-mechanical treatment process [21]. Xiao et al. found that the addition of Co can suppress spinodal decomposition and stimulate precipitation and then leads to higher mechanical property and electrical conductivity [22]. In fact, whether there exists spinodal decomposition in Cu-Ni-Si alloy during aging or not remains to be resolved. It is well know that the reflection spots resulted from the spinodal decomposition presents in the soft modulus directions of materials in order to accommodate the strain. In the study by Xiao et al., the four weak spots around each 200_Cu_ spots of fcc matrix in the [001]_Cu_ diffraction pattern are paralleled to <110> directions. In fact, the soft modulus directions of face-centered cubic material should be <001> directions. Monzen et al. also did not explain why the addition of Co can decrease the interprecipitate spacing.

In this study, the effects of minor addition of Co on the properties and microstructure of Cu-Ni-Si-P-Mg alloy are investigated in detail. We found that the addition of Co to the alloy enhances the hardness without decreasing its electrical conductivity. The reason for the increase in the hardness for the Co-contain Cu-Ni-Si-P-Mg alloys after aging was also discussed.

## 2. Materials and Methods

Cu-Ni-Co-Si-P-Mg alloys with different composition (wt.%) were prepared by vacuum induction melting and casting in an iron mold, with the raw materials of electrolytic cathode pure copper bulk (99.95%), pure nickel sheet (99.9%), pure cobalt bulk (99.9%), pure silicon granule (99.99%), Cu-10 wt.% P master alloy, pure magnesium sheet (99.99%). Ingots were surface milled and then homogenized at 850 °C for 24 h, subsequently hot rolled at 800 °C to 4 mm thick sheets. The hot rolled samples were then solution treated at 950 °C for 2 h before they were water quenched. Subsequently, two kinds of thermo-mechanical treatments processes are conducted (listed in Table 1). The thickness reduction of cold rolling for five alloy plates (C0 to C4) is 60%, 57%, 47%, 55% and 46%, respectively. The nominal chemical compositions of Cu-Ni-Co-Si-P-Mg (wt.%) alloys were listed in Table 2. 

The micro-hardness measurements were conducted on a HXS-1000A microscopic Vickers hardness tester (Shangguang, Shanghai, China) with a load of 300 g and a holding time of 15 s. Electrical conductivity was measured by a FQR7501A eddy current conductivity meter (MKY, Beijing, China) at room temperature. At least five measurements were taken for each data point in both of the cases, i.e., hardness and conductivity. The conductivity was measured and evaluated according to the international annealing copper standard (IACS, 100% IACS = 58 MS·m^−1^ = 1.7241 μΩ·cm). Samples for transmission electron microscopy (TEM) were thinned to a thickness of 0.1 mm by paper grinding (2500 grit). Disks of 3 mm in diameter were punch out from these samples. These disks were further thinned for electron transparency using a DJ-2000 twin jet electropolisher (Dervee instruments Co., Suzhou, China) with a solution of 25% HNO_3_ + 75% CH_3_OH at −30 °C. Transmission electron microscopy (TEM) and scanning transmission electron microscopy-high angle annular dark field (STEM-HAADF) observations were both performed using a Talos F200x microscope (FEI, Portland, OR, USA) operated at 300 kV. Energy Dispersive X-ray Spectroscopy (EDS) was performed with a X-Max SDD EDS system (4 SDD, Oxford instruments, Oxford, England) and a probe size of 0.2 nm.

## 3. Results and Discussion

### 3.1. Effects of Co Addition on Hardness and Electrical Conductivity of Cu-Ni-Si-P-Mg Alloys

Figure 1 shows the microhardness and electrical conductivity of five as-quenched Cu-Ni-Co-Si-P-Mg alloys with different Co content as a function of aging time at 450 and 500 °C. The hardness and electrical conductivity values of the five alloys increase rapidly during the initial aging. Fast precipitate nucleation leads to the rapid increase of hardness and electrical conductivity after 30 min aging. Subsequently, the hardness values increase with the increasing amount and size of precipitates until the peak values. For C0 and C1 alloys, as shown in Figure 1a, the hardness reaches the peak value of 250 and 258 HV after aging at 450 °C for 4 and 6 h, respectively, while for the C2, C3 and C4 alloy the hardness reaches the peak value of 230, 254 and 249 HV after about 12 h aging and the peak hardness values are less than those of the C0 and C1 alloys. This indicates the partial substitution of Co for Ni delays significantly the age-hardening response of the alloy. It is beneficial to industrial production of Cu-Ni-Co-Si alloys strip because large heavy winded roll of alloy strip need enough time to heat penetrate. However, long time heat treatment will result in over-aging of these alloys. Figure 1b shows that the five alloys almost have the same electrical conductivity in the same aging condition. It suggested that the Co substitution for Ni scarcely affects the electrical conductivity over the whole aging for the as-quenched alloys. Figure 1c illustrates that the hardness values of the C0 and C1 alloys dramatically increase to the peak of 235 and 251 HV after aging at 500 °C for 1 and 1.5 h, respectively, while the hardness values of the C2, C3 and C4 alloys reach the peak values of 235, 257 and 247 HV after aging for 1.5, 3 and 3 h, respectively. The hardness values of the C0 and C1 alloys sharply decrease with prolonged aging time due to over-aging. In addition, the larger the Co content in the alloys is, the smaller the hardness decrement with over-aging time is. This suggests the substitution of Co for Ni can not only improve the hardness, but also can lower the coarsening rating of the precipitates in these alloys. Figure 1d shows that the electrical conductivities of the C0 alloy after aging at 500 °C are higher than those of counterpart alloys after aging at 450 °C. The C1, C3 and C4 alloys except C2 almost have the same electrical conductivities as aging at 500 °C for the same time. 

Figure 2 shows the microhardness and electrical conductivity of the five cold-rolled Cu-Ni-Co-Si-P-Mg alloys with different Co content as a function of aging time at 450 and 500 °C. As shown in Figure 2a, the hardness of the C0 and C1 alloys reach the peak value of 249 and 281 HV after aging at 450 °C for 1.5 h, respectively, while the hardness of the C2, C3 and C4 alloys reach the peak value of 279, 283 and 276 HV after aging for 3, 3 and 4 h, respectively. The peak hardness values of the four Co-containing alloys are much higher than that of the C0 alloy. Comparing to the as-quenched alloys, the time to reach the hardness peak of the five alloys is shorten greatly. In addition, the time to reach the hardness peak is delayed with the increase of the Co content in the alloys. Figure 2b shows that the electrical conductivities of the C0 and C1 alloys are higher than those of the C2, C3 and C4 alloys over the whole aging. A shown in Figure 2c, the peak hardness values of 258, 290, 280, 276 and 286 HV for the C0~C4 alloys are obtained after aging at 500 °C for about 0.5, 0.5, 0.5, 1.5 and 1.5 h, respectively. It reveals that cold rolling can significantly accelerate the precipitation process. Subsequently, the hardness values of these alloys began to decrease and then reach 166, 193, 220, 225 and 215 HV after aging for 16 h. The decrement values in hardness for the C0 and C1 alloys are larger than those for the C2, C3 and C4 alloys. It confirmed again that the Co substitution for Ni can improve the hardness of Cu-Ni-Si alloy and postpone the time to reach the peak hardness of aging. However, the electrical conductivities values of the C0 and C1 are higher than those of C2, C3 and C4 alloys in the same aging time condition (Figure 2d). After aging for 2 h, the electrical conductivities of the C0~C4 alloys reach 44.8, 43.7, 39.5, 40.0 and 40.8%IACS, respectively. The increase of the electrical conductivity and hardness can be attributed to the removal of dissolved solute atoms from the copper matrix by the formation of precipitates. The cold rolling before aging can increase the density of defects like dislocations and vacancies in the alloys. The dislocations and vacancies act as nucleation sites for further aging and then enhance the rate of precipitation and growth.

In summary, according the aging behavior according to Figure 1 and Figure 2, the alloys can be divided into two groups. C0 and C1 with Co/Ni weight ratio of 0.27 have the same aging behavior and thus can be classified in one group. C2-C4 alloys with more than 0.6 wt.% Co (Co/Ni weight ratio >0.27) can be classified in another one. In addition, C1 alloy can obtain higher peak hardness than Cu-2.8Ni-0.65Si -0.1P-0.05Mg alloy without Co and almost the same electrical conductivity. Addition of Co (more than 0.6 wt.%, or Co/Ni weight ratio >0.27) can postpone the time of peak hardness during aging. Compared with the studied alloys aged at 450 °C, the classification for these alloys aged at 500 °C is more obvious. The as-quenched Cu-Ni-Co-Si-P-Mg alloys with Co have the higher hardness than Cu-Ni-Si-P-Mg alloy and a small decrease in hardness after a prolonged aging. It revealed that the alloy containing Co has the excellent thermal stability. When the Co content is larger than 0.6 (Co/Ni weight ratio >0.27), the effect of Co on improving hardness was impaired and the electrical conductivity decreased with the increase of Co content instead.

### 3.2. Effect of Co Addition on Microstructure of Cu-Ni-Si-P-Mg Alloys

The high strength and electrical conductivity of precipitation hardening Cu-Ni-Si system alloys are mainly controlled by the chemical composition, morphology, size and distribution of nano-sized precipitates. We took the as-quenched C0, C1 and C4 alloys as examples to illustrate the effect of Co addition on the feature of the precipitates during aging at 500 °C. After aging at 500 °C for 0.5 h, TEM micrographs of C0, C1 and C4 alloys and corresponding selected area electron diffraction (SAED) patterns are presented in Figure 3. Figure 3a,b show the precipitates with an average size of 9–10 nm in diameter and 3–4 nm in thickness uniformly distributed in the matrix. Figure 3a illustrates a high resolution transmission electron microscopy (HREM) image of precipitates in the C0 alloy aged for 0.5 h and the corresponding fast Fourier transform (FFT) pattern in the inset. The indexing result of the pattern confirms that these precipitates are mainly orthorhombic δ-Ni_2_Si phase (a = 0.708 nm, b = 0.502 nm, c = 0.371 nm) [3,6,23]. Figure 3b show a bright field (BF) TEM image of two mutually perpendicular disc-like δ-Ni_2_Si variants lying on the {110}_Cu_ planes align in the two <110> _Cu_ directions and the corresponding [001]_Cu_ selected area electron diffraction (SAED) pattern in the inset. It can be deduced from Figure 3b that there exist six δ-Ni_2_Si variants in this alloy. The crystallographic orientation relationships (ORs) between the Cu matrix and precipitates can be concluded from the above diffraction analysis: (001)_Cu_ //(001)_δ_, [110]_Cu_ //[010] _δ_. In addition, it can be found that there exists obvious dark strain contrast around two disc sides of these δ-Ni_2_Si precipitates. It suggests that these nano-sized δ-Ni_2_Si precipitates are fully coherent with Cu matrix. Figure 3c,d reveal that the size of precipitates decrease to about 7–8 nm in diameter and 3–4 nm in thickness for the C1 alloy, compared with that observed in the C0 alloy. The SAED patterns of precipitates in the C1 alloy are the same as those in the C0 alloy. Figure 3e,f show that the average size of precipitates in the C4 alloy is the smallest among the three alloys, which diameter and thickness are 5–6 and 2–3 nm, respectively. It indicates that the addition of Co can reduce the size of precipitates. The larger the amount of Co in the alloys is, the smaller the precipitates formed is. 

After further aging at 500 °C for 4 h, the precipitates in the Co-free C0 alloy obviously coarsened (Figure 4a), which average diameter and thickness increase to about 17–18 and 4–5 nm, respectively. Figure 4a also reveals clearly the other disc-like δ-Ni_2_Si variants in the C0 alloy. However, the average diameter and thickness of the precipitates in the C1 alloy are about 13 and 4 nm as shown in Figure 4b–d. For the C4 alloy, the STEM HAADF images in Figure 5 show that these precipitates are about 11 nm in diameter and 4 nm in thickness. The elemental mapping images (Figure 5a–e) demonstrate that these precipitates are associated with Cu, Co, Ni and Si. The elemental mapping microanalyses of X –ray energy dispersive spectrum (EDS) reveal clearly the poorness of Cu and richness of Ni, Co and Si in the disc-like particles. However, Mg seem to exist in the matrix in the form of the solid solution atom. Figure 5f gives the chemical composition of the particle (labeled by “Area #1” in Figure 5a) as follows (at.%): 4.98 Co, 3.64 Ni, 4.51 Si, 1.26 P and 0.25 Mg. Its atomic ratio of (Co + Ni)/Si in this particle is about 1.9 and is close to stoichiometry Ni/Si ratio of 2 in δ-Ni_2_Si precipitates. Therefore, the precipitates in the Co-containing C1 and C4 alloys are (Ni, Co)_2_Si phase. 

After aging at 500 °C for 16 h, the average size of δ-Ni_2_Si precipitates in the C0 alloy reaches about 25–30 nm in diameter and 4–5 nm as shown in Figure 6a,b. However, it can be found from Figure 6a that there exist the strain contrasts in matrix around the δ-Ni_2_Si precipitates. It suggests that the precipitates keep still coherency with the Cu matrix. For the C1 alloys, Figure 6c,d reveal the existence of three δ-Ni_2_Si variants viewed in the <111>_Cu_ directions. The average size of these δ-(Ni, Co)_2_Si precipitates are measured to be about 20 nm in diameter and 5 nm in thickness. For the C4 alloy, The HAADF image (Figure 6e) and HREM image (Figure 6f) show that the average size of these δ-(Ni, Co)_2_Si precipitates has increased to about 15 nm in diameter and 4 nm in thickness.

It is worth noting that a rod-like or plate-like phase was found in the C0 alloy. This phase is about 150 nm in length (marked by red arrow in Figure 6a,b), which is obviously larger than the δ-Ni_2_Si precipitate phase. The habit planes of the coarse precipitate phase parallel to the {100}_Cu_ planes as shown in the inset of Figure 6a. Viewed in the [001]_Cu_ direction, two mutual perpendicular variants can be found in Figure 6a,b. The habit planes of the two variants have a 45°angle with those of δ-Ni_2_Si. This precipitate phase can also be found in the C0 alloy aged at 500 °C for different time. This phase was also marked by red arrow in Figure 4a and Figure 7. The HREM image and the corresponding FFT images shown in Figure 7 and its insets reveal a different diffraction patterns with δ-Ni_2_Si and a different orientation relationship with the Cu matrix. Figure 8a shows that this new phase appears to be plate-like or needle-like. Its corresponding SAED pattern is marked by yellow dash-line in Figure 8b. The diffraction pattern of the P-enriched phase is different from the ones of Ni-P phase in Cu-Ni-Si-P alloy reported by Kim [4] and from the ones of Ni-P phase in Cu-Ni-P alloys reported in literature [24,25,26]. Unfortunately, it is difficult to determine the structure of this new phase due to the limited information on SAED patterns in present study. Figure 8c and d illustrate the HAADF image and EDS microanalysis result of the new phase, respectively. This new phase (labeled by “Area #1” in Figure 8c) contains 32.46 Cu, 42.25 Ni, 16.34 P, 3.84 Si and 5.11 Mg (at. %), where Ni/(P+Si) ratio is close to 2. Figure 9 shows that the EDS element mapping images of coarse plate-like precipitates in Figure 8 for C0 alloy aged at 500 °C for 16 h. Cu, Ni and P enriched in the coarse plate-like precipitates. Mg seems to be also rich in these plate-like precipitates.

For the precipitation-hardened alloys, precipitation-hardening is one of the most important and effective strengthening methods. It is well known that precipitation-hardening requires an optimal combination of precipitate size and number density to yield the maximum combination of properties, such as yield strength, hardness, and electrical conductivity for copper alloy. In order to obtain a deep insight into the correlation among the composition, microstructure and property in Cu-Ni-Si system copper alloys, it is necessary to understand the strengthening mechanisms of the nanoscale precipitates. According to the interaction mechanism between the precipitates and the gliding dislocations, there are two strengthening mechanisms, namely dislocation shearing and Orowan dislocation bypassing mechanisms [27,28]. When the dislocations cut through precipitates, the strengthening effect Δ*σ_coherency_* of precipitates is roughly proportional to the square root of total volume fraction f and mean size d of precipitates, which can be calculated by the Equation (1) [29]: (1)$$\Delta \sigma_{coherency}=M\alpha_{E}G\varepsilon_{c}^{3/2}{\left(\frac{df}{b}\right)}^{1/2}$$
where *G* is the shear modulus of matrix, *M* is the mean matrix orientation factor, *b* is the magnitude of the matrix Burgers vector, *ε_c_* = 2/3(Δ*a/a*) is the constrained lattice parameter mismatch, with Δ*a/a* as the lattice parameter mismatch at room temperature [4]; *α_E_* = 2.6 for face-centered cubic (fcc) metals. When the dislocations bypass the precipitates, the strengthening effect Δ*σ_Orowan_* of precipitates is roughly proportional to the square root of total volume *f* of precipitates and inverse proportional to mean precipitate size *d*. It is given by Equation (2) [29,30]:(2)$$\Delta \sigma_{Orowan}=M\left(\frac{0.4Gb}{\pi \sqrt{1-\nu }\left[{\left(\frac{3\pi }{4f}\right)}^{1/2}-1.64\right]r}\right)\ln\left(\frac{2\bar{r}}{b}\right)$$
where *ν* is the matrix Poisson’s ratio; $\bar{r}$ is r is the mean radius of a circular cross-section in a random plane for a spherical precipitate, $\bar{r}=\sqrt{2/3}r$, where r is the mean radius of the precipitates. Therefore, supposing that the total volume of precipitates was given, the strengthening contribution depends on the mean size of precipitates. In the present studies, as shown in Figure 1c and Figure 3, the hardness of the under-aged Co-containing C1-C4 alloys are lower than those of the C0 alloy because of the smaller mean size of precipitates when aging at 500 °C for the same time (up to 30 min). In condition of under-aging, the dislocation gliding is governed by shearing mechanism. The strengthening contribution increases with the increase in the mean size of precipitates. When the mean size of precipitates is larger than the critical size, the dislocation gliding is dominated by the Orowan process and the strengthening contribution will decrease with the increase in the mean size of precipitates. The hardness values of the C0–C4 alloys began to decrease when aging at 500 °C from 1, 1.5, 1.5, 3 and 3 h, respectively. As shown in Figure 4, Figure 5 and Figure 6, the average size of precipitates in the C0 alloy is much larger than the Co-containing alloys. Therefore, the hardness values of the C0 alloy is less than those of the Co-containing alloys after over-aging. The dependence of hardness and precipitate size on the Co content in the alloy is more obvious when aged at 450 °C (Figure 1a). 

Based on the observation of the microstructure evolution for the studied alloys, it can be found that the suitable substitution of Co for Ni can not only improve the hardness of the alloy, but also lower the coarsening rate of the precipitates in this alloy. This improvement in hardness demonstrates the significant effect of Co on the mechanical properties, which may be attributed to the precipitate size and their kinetics. It was reported that the impurity diffusion of Co (8.4 × 10^−5^ m^2^ s^−1^) in Cu [31,32] is slower by about an order of magnitude than that of Ni (2.7 × 10^−4^ m^2^ s^−1^) in Cu [33,34]. The smaller diffusion coefficient of Co in Cu matrix leads to the slower growth rate of δ-(Ni, Co)_2_Si precipitates. Therefore, when the mean sizes of precipitates in the Co-containing aged alloy are smaller than that in the Co-free aged alloy. The higher the Co content in these alloys is, the small the mean size of precipitates is. In addition, the P-enriched precipitate phase is obviously easier to coarsen than the δ-Ni_2_Si and δ-(Ni, Co)_2_Si. To some extent, it results in a greater increase in hardness during under-aging and greater decrement for the Co-free alloy during over-aging. The optimal addition amounts of Ni and Co are 1.6 wt.% Ni and 1.2% Co according to the hardness and electrical conductivities of the aged alloys in the present study. 

## 4. Conclusions

According the aging behavior, the studied alloys can be divided into two groups. Cu-2.8Ni-0.65Si-0.1P-0.05Mg and Cu-2.2Ni-0.6Co-0.65Si-0.1P-0.05Mg with Co/Ni weight ratio of 0.27 have the same aging behavior, but Cu-2.2Ni-0.6Co-0.65Si-0.1P-0.05Mg can obtain higher peak hardness than the former and almost the same electrical conductivity. For another group of Cu-(2.8 − x)Ni-xCo-0.65Si-0.1P-0.05Mg (*x* ≥ 1 or Co/Ni weight ratio ≥0.56) alloys, the aging time to reach the peak hardness can be postponed during aging. However, its electrical conductivity decreases slightly with the increase of Co content.The addition of Co can reduce the size of precipitates. The larger the amount of Co in the alloys is, the smaller the precipitates formed is. Meanwhile, the addition of Co in the Cu-Ni-Si-P-Mg alloy can also inhibit the formation of P-enriched phase in the Co-containing alloys during aging. The P-enriched phase has a definite orientation relationship with the matrix, which habit plane is parallel to the {001} planes of Cu matrix. The P-enriched precipitate phase is obviously easier to coarsen than the δ-Ni_2_Si and δ-(Ni, Co)_2_Si, which results in a greater decrement in hardness for Co-free alloy during over-aging.The optimal addition amounts of Ni and Co are 1.6 wt.% Ni and 1.2% Co. The hardness and electrical conductivity of the as-quenched Cu-1.6Ni-1.2Co-0.65Si-0.1P-0.05Mg alloy increase to 257 HV and 38.7 %IACS after aging at 500 °C for 3 h, respectively.

## Figures and Tables

**Figure 1 materials-14-00368-f001:**
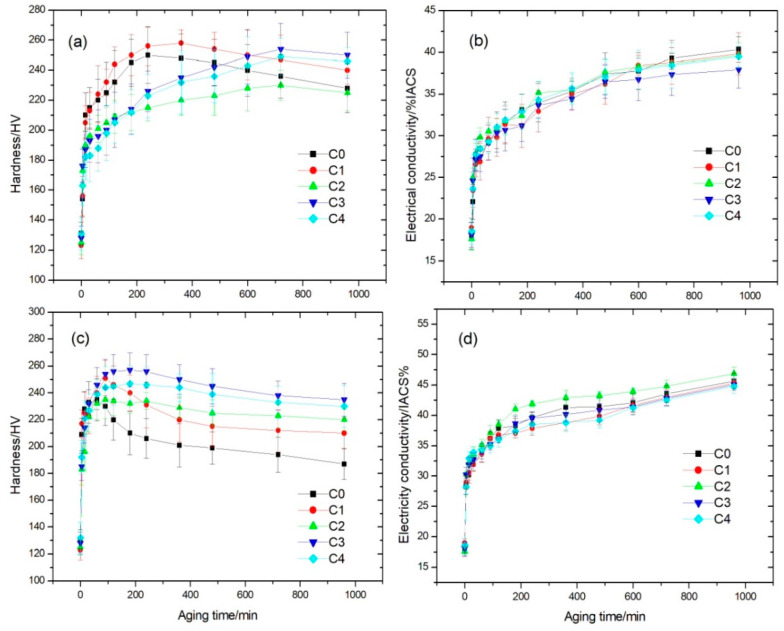
Hardness and electrical conductivity curve of as-quenched Cu-Ni-Co-Si-P-Mg alloys as a function of aging time at different aging temperature (**a**) hardness, aging at 450 °C, (**b**) electrical conductivity, aging at 450 °C, (**c**) hardness, aging at 500 °C, (**d**) electrical conductivity, aging at 500 °C.

**Figure 2 materials-14-00368-f002:**
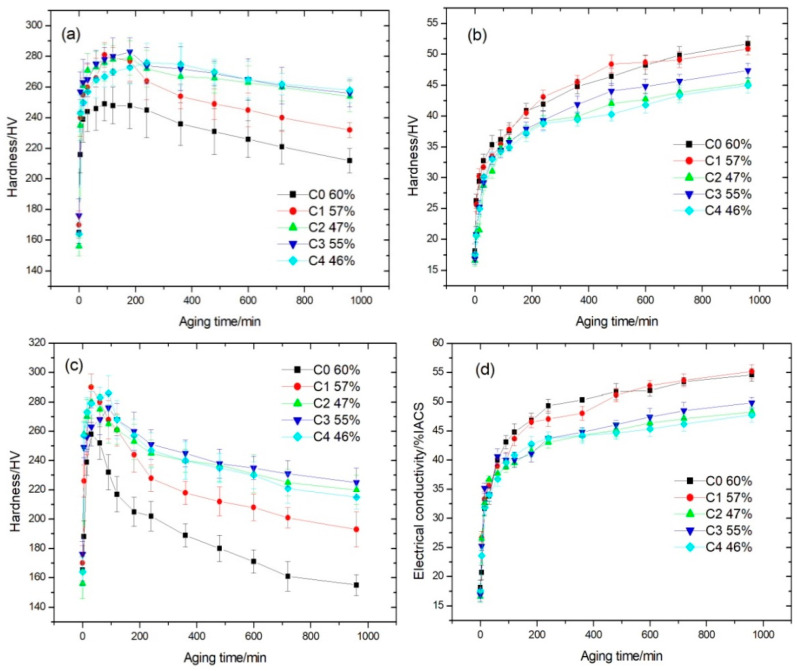
Hardness and electrical conductivity curve of cold rolled Cu-Ni-Co-Si-P-Mg alloys as a function of aging time at different aging temperature. (**a**) hardness, aging at 450 °C, (**b**) electrical conductivity, aging at 450 °C, (**c**) hardness, aging at 500 °C, (**d**) electrical conductivity, aging at 500 °C.

**Figure 3 materials-14-00368-f003:**
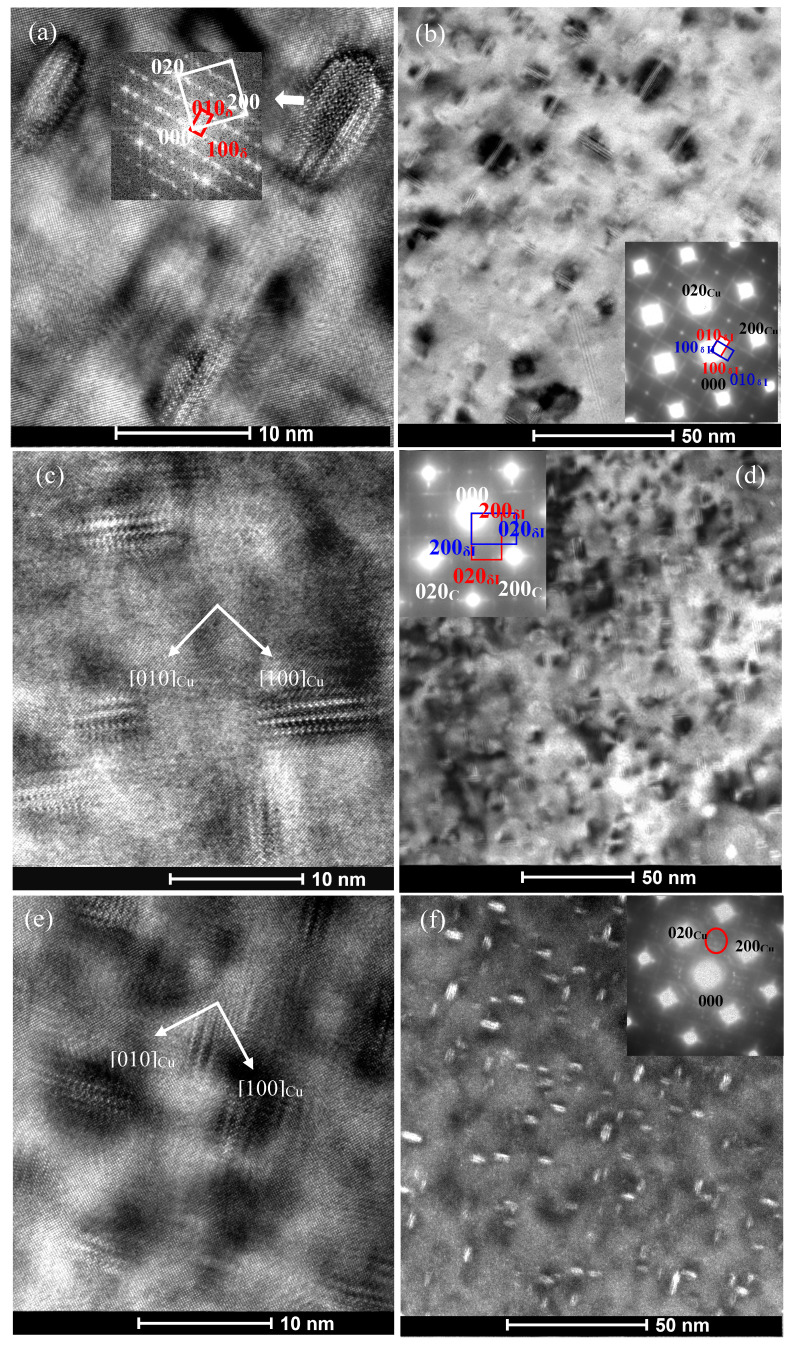
TEM images obtained from the C0, C1 and C4 alloys aged at 500 °C for 0.5 h (**a**) HREM image of C0 alloy along [001]_Cu_ zone axis and corresponding FFT pattern in the inset, (**b**) Dark field (DF) image of C0 alloy and corresponding [001]_Cu_ pattern in the inset, (**c**) HREM image of C1 alloy along [001]_Cu_ zone axis, (**d**) DF image of C1 alloy and corresponding [001]_Cu_ pattern in the inset, (**e**) HREM image of C4 alloy along [001]_Cu_ zone axis, (**f**) DF image of C4 alloy and corresponding [001]_Cu_ pattern in the inset. The DF image was taken using precipitate (020) reflection marked by red circle in the inset. Red and blue rectangles refer to the diffraction pattern of two δ-Ni_2_Si variants and the same below.

**Figure 4 materials-14-00368-f004:**
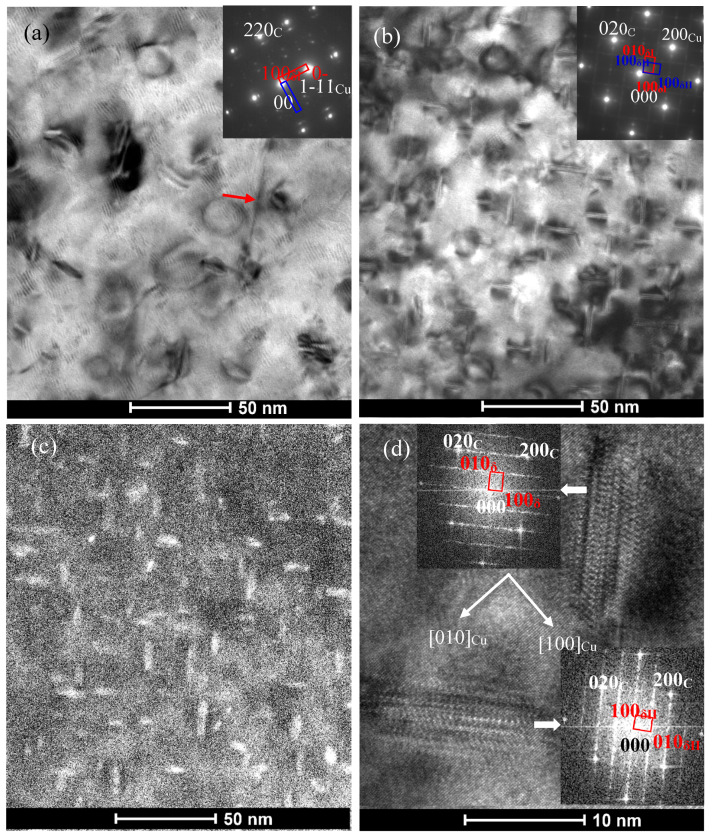
TEM images obtained from the C0 and C1 alloys aged at 500 °C for 4 h (**a**) BF image of C0 alloy and corresponding [−112]_Cu_ SAED pattern in the inset, (**b**) BF image of C1 alloy and corresponding [001]_Cu_ SAED pattern in inset. (**c**) DF image of C1 alloy along [001]_Cu_ zone axis, (**d**) HREM image of C1 alloy along [001]_Cu_ zone axis and FFT patterns of two precipitate variants in the inset. Red arrow refers to the P-rich phase.

**Figure 5 materials-14-00368-f005:**
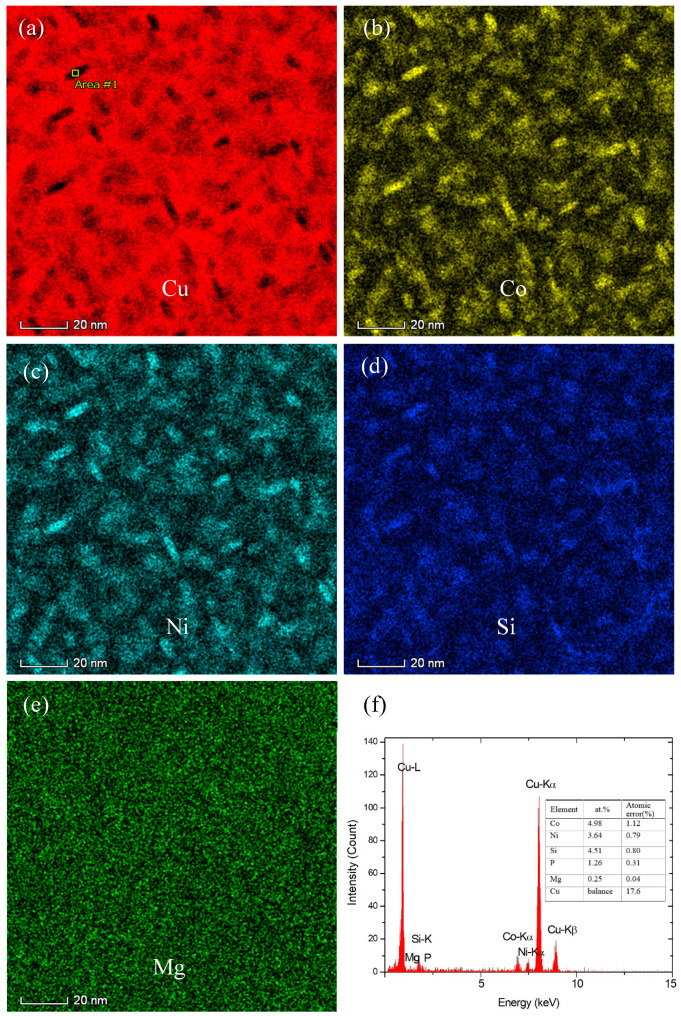
STEM elemental mapping images of the C4 alloy aged at 500 °C for 4 h. (**a**) Cu, (**b**) Co, (**c**) Ni, (**d**) Si, (**e**) Mg, (**f**) EDS result of a precipitate particle (labeled by “Area #1” in (**a**).

**Figure 6 materials-14-00368-f006:**
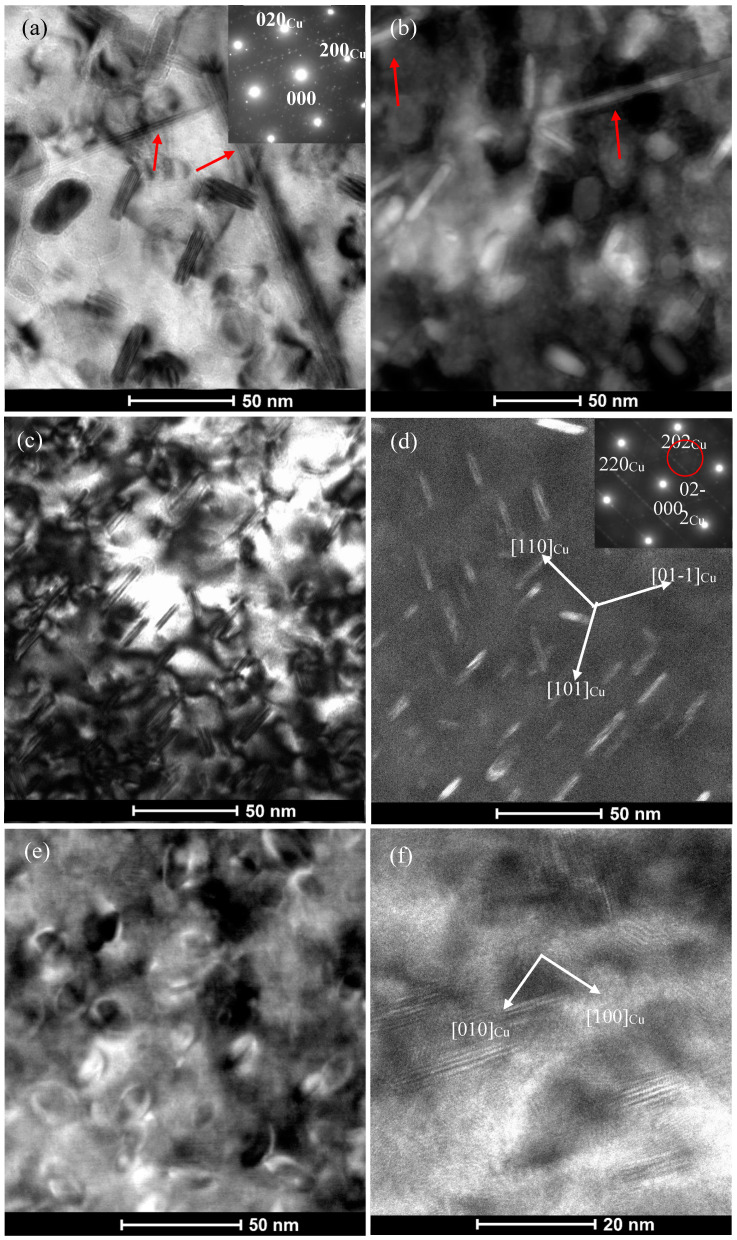
TEM images obtained from the C0, C1 and C4 alloys aged at 500 °C for 16 h (**a**) BF image of C0 alloy along [001]_Cu_ zone axis and corresponding SAED in the inset, (**b**) HAADF image of C0 alloy along [001]_Cu_ zone axis, (**c**) BF image of C1 alloy along [−111]_Cu_ zone axis, (**d**) DF image of C1 alloy and corresponding [−111]_Cu_ pattern in the inset. DF image was taken using the reflections (marked by red circle) of precipitates. (**e**) HAADF image of C4 alloy, (**f**) HREM image of C4 alloy along [001]_Cu_ zone axis. Red arrows in (a) and (b) refer to the P-rich phase.

**Figure 7 materials-14-00368-f007:**
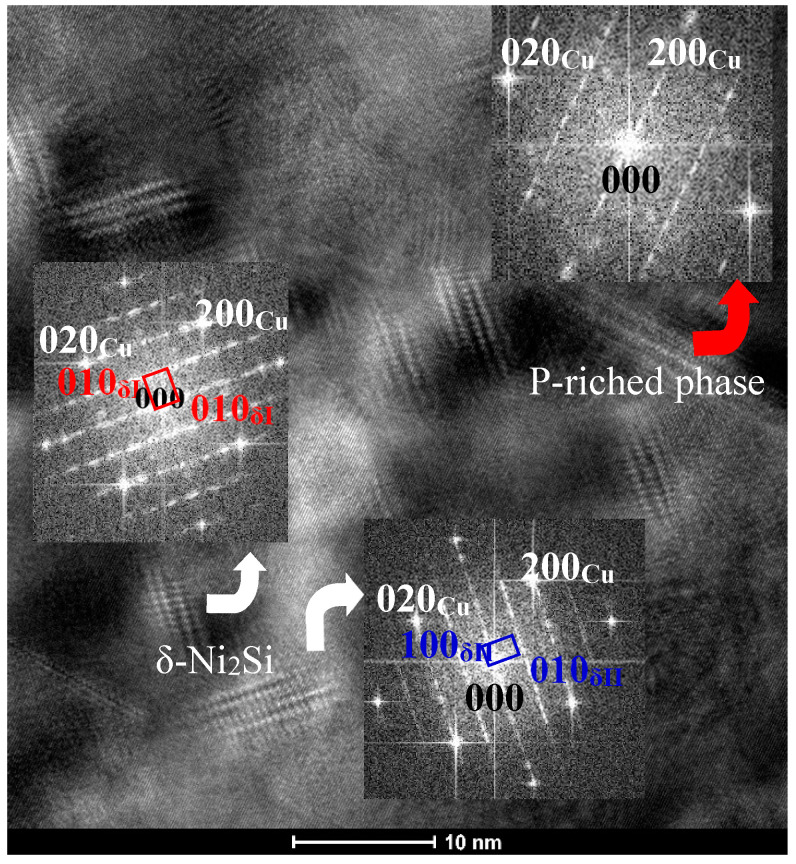
HREM image of C0 alloy aged at 500 °C for 15 min and corresponding FFT patterns of precipitates.

**Figure 8 materials-14-00368-f008:**
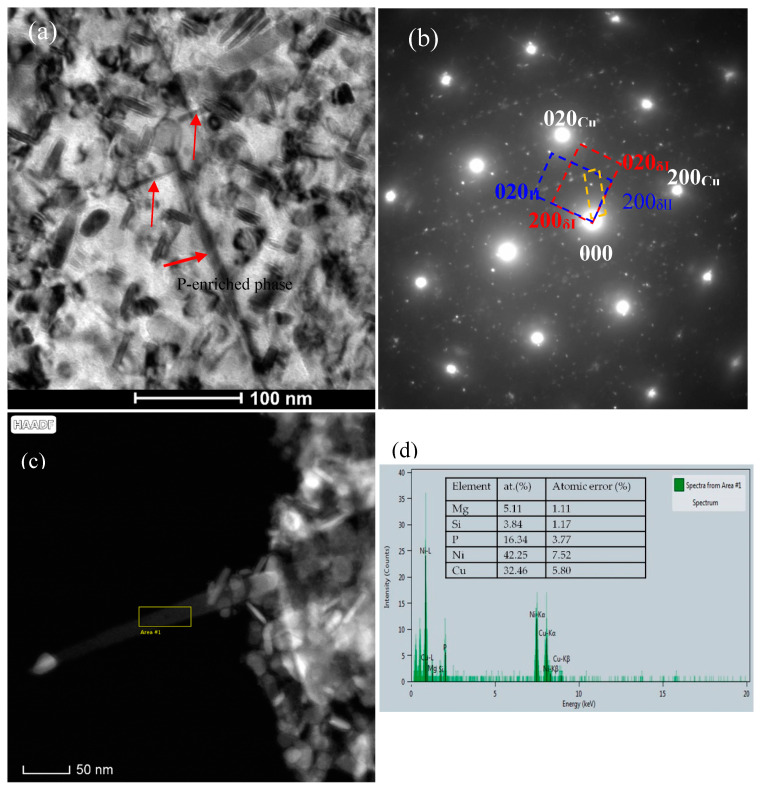
TEM and STEM images of C0 alloy aged at 500 °C for 16 h (**a**) [001]_Cu_ BF image, (**b**) corresponding [001]_Cu_ SAED pattern of (**a**), two sets of pattern (marked by red and blue dash line) belong to the two δ-Ni_2_Si variants, one (marked by yellow dash-line) belongs to the P-enriched phase. (**c**) HAADF image of needle-like P-enriched phase and δ-Ni_2_Si variants, (**d**) EDS analysis result of P-enriched phase (labeled by “Area #1” in (**c**). Red arrow in (a) refer to the P-rich phase.

**Figure 9 materials-14-00368-f009:**
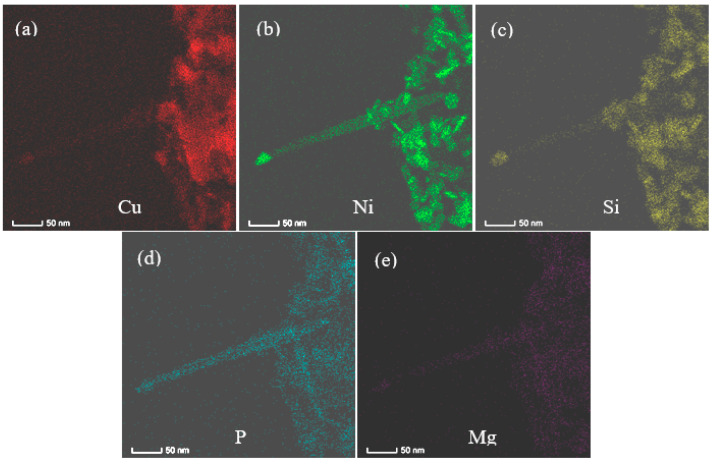
EDS element mapping images of plate-like precipitates of Figure 8 in C0 alloy aged at 500 °C for 16 h. (**a**) Cu, (**b**) Ni, (**c**) Si, (**d**) P, (**e**) Mg.

**Table 1 materials-14-00368-t001:** Two kinds of thermo-mechanical treatment processes.

Process	Route
I	Aged at 400, 450 and 500 °C for different time, respectively.
II	Cold rolled and then aged at 450 and 500 °C for different times, respectively.

**Table 2 materials-14-00368-t002:** Nominal Chemical compositions of Cu-Ni-Co-Si-P-Mg alloys (wt.%).

Alloy No.	Ni	Co	Si	P	Mg	Cu
C0	2.8	0	0.65	0.1	0.05	balance
C1	2.2	0.6	0.65	0.1	0.05	balance
C2	1.8	1.0	0.65	0.1	0.05	balance
C3	1.6	1.2	0.65	0.1	0.05	balance
C4	1.2	1.6	0.65	0.1	0.05	balance

## Data Availability

Data sharing is not applicable to this article.

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
