# Peer review of "Effects of Co Addition on the Properties and Microstructure of Cu-Ni-Si-P-Mg Alloys"

_materials, 2021, doi:10.3390/ma14020368_

Round 1

Reviewer 1 Report

The authors proposed a research on the Co  addition and the relative impact on the Cu-Ni-Si-P-Mg alloys.

The paper is interesting but need some rivision before pubblication:

  • The authors should insert in the abstract a short description of the most important results;
  • The multi references (for example [1-5], [3,7, 12-18] etc are not a good practice. The authore shoud be more specific on the single reference.
  • The author should improve the description of the experimental setup. 
  • How many tests were used in the experimentation. How many replication? A solution could be a table summarizing these information

Author Response

Reviewer #1: The paper is interesting but need some revision before publication.

Comment 1:The authors should insert in the abstract a short description of the most important results.

Comment 2: The multi references (for example [1-5], [3,7, 12-18] etc are not a good practice. The authore shoud be more specific on the single reference.

Comment 3: Author should improve the description of the experimental setup. 

Comment 4:How many tests were used in the experimentation. How many replication? A solution could be a table summarizing these information.

  1. Response to comment 1

Thanks for your suggestion. We have picked up the important results in the conclusions part.

  1. Response to comment 2

We have marked the references and specified each reference according to your request. Please see the line 34-39 page1,“Therefore, some attempts have been made to improve the performance of Cu-Ni-Si alloys by increasing their Ni and Si contents and/or adding alloying elements such as Cr [3], Ti [12,13,14], V[15], Al[16], P[17], and Zr [18].”

3 Response to comment 3

We have revised the part “materials and method” and corrected some faults about the raw materials and listed the two kinds of heat treatment processes in the Table 1.

Cu-Ni-Co-Si-P-Mg alloys with different composition (wt. %) were prepared by vacuum induction melting and casting in an iron mold, with the raw materials of electrolytic cathode pure copper bulk (99.95%), pure nickel sheet (99.9%), pure Cobalt bulk (99.9%),pure silcon granule (99.99%),Cu-10 wt. % P master alloy, pure magnesium sheet (99.99%). Ingots were surface milled and then homogenized at 850oC for 24h, subsequently hot rolled at 800 oC to 4 mm thick sheets. The hot rolled samples were then solution treated at 950 °C for 2 h before they were water quenched. Subsequently, two kinds of thermo-mechanical treatments processes are conducted (listed in Table 1). The thickness reduction of cold rolling for five alloy plates (C0 to C4) is 60%, 57%, 47%, 55% and 46%, respectively. The nominal chemical compositions of Cu-Ni-Co-Si-P-Mg (wt. %) alloys were listed in Table 2.

Table 1 Two kinds of thermo-mechanical treatment processes

Process

Route

I

Aged at 400°C, 450°C and 500°C for different time, respectively.

II

Cold rolled and then aged at 450°C and 500°C for different times, respectively.

Table 2. Nominal Chemical compositions of Cu-Ni-Co-Si-P-Mg alloys (wt. %).

Alloy No.

Ni

Co

Si

P

Mg

Cu

C0

2.8

0

0.65

0.1

0.05

balance

C1

2.2

0.6

0.65

0.1

0.05

balance

C2

1.8

1.0

0.65

0.1

0.05

balance

C3

1.6

1.2

0.65

0.1

0.05

balance

C4

1.2

1.6

0.65

0.1

0.05

balance

The micro-hardness measurements were conducted on a HXS-1000A microscopic Vickers hardness tester with a load of 300 g and a holding time of 15 seconds. Electrical conductivity was measured by FQR7501A eddy current conductivity meter at room temperature. At least five measurements were taken for each data point in both of the cases, i.e. hardness and conductivity. The conductivity was measured and evaluated according to the international annealing copper standard (IACS, 100% IACS = 58 MS m-1 = 1.7241 μΩ cm). Samples for transmission electron microscopy (TEM) were thinned to a thickness of 0.1 mm by paper grinding (2500 grit). Disks of 3 mm in diameter were punch out from these samples. These disks were further thinned for electron transparency using a DJ-2000 twin jet electropolisher with a solution of 25% HNO3 + 75% CH3OH at -30 °C. Transmission electron microscopy (TEM) and scanning transmission electron microscopy-high angle annular dark field (STEM-HAADF) observations were both performed using a Talos F200x microscope operated at 300 kV. Energy Dispersive X-ray Spectroscopy (EDS) was performed with a X-Max SDD EDS system (4 SDD) and a probe size of 0.2 nm.

4 Response to comment 4

In the manuscript, we have mentioned the number of micro-hardness test and electrical conductivity measurement. See the above part.

Reviewer 2 Report

Review of «Effects of Co Addition on Properties and Microstructure of Cu-Ni-Si-P-Mg Alloys» - Materials 1034048

1. Summary. The paper deals with the fabrication of Cu-based alloys for electric and electronic industry. In particular, the paper deals with the analysis of the mechanical, electrical and microstructural features of five alloys, with varying Co and Ni content. The alloys were studied in the as-cast state, after a series of annealing and in the condition of mechanical deformation, obtained by cold rolling. The mechanical properties were studied by microhardness studies, while the electrical properties were assessed by conductivity studies. Transmission electron investigations were carried out to provide an interpretation key for the mechanical and electrical behaviour of the alloys. The subject of the paper is compatible with the topics covered by MDPI «Materials». The work is well organized, even if some major modifications are required before publication. The present document describes the required modifications to be brought to the manuscript to improve it until the final version. Three main subject are covered, that is (1) General comments; (2) Punctual comments, and (3) Use of English.

2. General comments. The paper is organized in a logic way. The performed characterizations are consistent with the given explanation. Data are presented in a well-organized way, but some improvements are needed, as expalined in the following session. In any case, the paper is presenting TEM analyses for just one annealing condition, even if for 4 different times.

3. Punctual comments.

A. The Abstract is missing one or two rows at the beginning, to introduce the background of this kind of alloys. The authors start in the middle of the action, which is somewhat abrupt for the reader.

B. page 2: Please, replace the word «Experimental» with «Materials and methods»

C. Page 2: Please, in the description of the alloy fabrication, provide the following details: Which are the details of the raw materials used to fabricate the alloys? size, chemical composition, impurities, etc. Which was the atmosphere in which the alloys were rolled? Why the thickness reduction was different for each chemical composition, and not the same for all the alloys? Which was the starting thickness for all the alloys? How the chemical composition of the alloys was checked, as the author speak just about nominal chemical composition? Which was the preparation surface for hardness (even if it should be addressed to as microhardness) and electrical conductivity?

D. Figure 1 and 2 are missing standard deviations, or any other measure of the dispersion of any data. Sometimes values are very close, so it is not possible to understand if the measure is consistent or not, and if the comparison is misleading or not.

E. page 2: «This indicates the partial substitution of Co for Ni delays significantly the age-hardening response of the alloy.» and also «This suggests the substitution of Co for Ni can not only improve the hardness of the alloy, but also can lower the coarsening rating of the precipitates in these alloys.»
The author should comment about the fact that it seems that there is some kind of threshold value of Co, or Co and Ni, as the alloys can divided into two groups, each one with a specific behavior: C0 and C1 on one side, and C2, C3 and C4 on the other side. This is specifically clear in Fig. 1a.

F. page 2 «It is well known that the conductivity depends on the rate of transport of solute atoms from the matrix to the precipitates.» it is not clear what the authors want to say here; it seems that conductivity is related to solute atoms movement, which is clearly not possible. Please, reorganize this sentence in a better way.

G. Please, provide indexing of Fig. 3b (inlet), Fig. 3f (inlet), Fig. 4(b) (inlet), Fig. 4(d) (inlets of FFT), Fig. 7 (inlets of FFT), Fig. 8(b). In some diffraction inlets, there are other patterns that need to be better assessed, for example in fig. 3b, 3f, 4a, 4b, 6a, 6f, 7b.

H. Please, provide a higher count spectrum for Fig.8(d), as there, the count number is very low.

I. Figure 5. Why is P not present? Which is the role of Mg and P in the alloy? Are they not impurities for the considered amounts?

4. Use of English. English needs to be improved, as there are expression that are not acceptable in current use. Here you are a few example; other ones can be found easily in the text.

page 1: «then leads to a higher mechanical property» should be replaced by «then leads to higher mechanical properties»

page 1: «did not explain that why the addition of Co» should be replaced by «did not explain why the addition of Co»

page 2: «small addition of Co on the propertied and microstructure» should be «properties», not «propertied»

page 2: «the addition of Co to the alloy enhances the hardness of the alloy» too many times alloy, here

page 2: «its electrical conductivity. And the reason for» «And» is not admissible after a full stop. In particular, this kind of mistake is rather frequent in the text. Please, correct it everywhere.

page 13: «Therefore, when the mean sizes of precipitates in the Co-containing aged alloy are smaller than that in the Co-free aged alloy. And the mean size of precipitates decreases with the increase of Co content in the studied alloys.» Please, reformulate the sentences, as they are not understandable in the present form.

Author Response

Reviewer #2:

  1. Summary.The paper deals with the fabrication of Cu-based alloys for electric and electronic industry. In particular, the paper deals with the analysis of the mechanical, electrical and microstructural features of five alloys, with varying Co and Ni content. The alloys were studied in the as-cast state, after a series of annealing and in the condition of mechanical deformation, obtained by cold rolling. The mechanical properties were studied by microhardness studies, while the electrical properties were assessed by conductivity studies. Transmission electron investigations were carried out to provide an interpretation key for the mechanical and electrical behaviour of the alloys. The subject of the paper is compatible with the topics covered by MDPI «Materials». The work is well organized, even if some major modifications are required before publication. The present document describes the required modifications to be brought to the manuscript to improve it until the final version. Three main subject are covered, that is (1) General comments; (2) Punctual comments, and (3) Use of English.
  2. General comments. The paper is organized in a logic way. The performed characterizations are consistent with the given explanation. Data are presented in a well-organized way, but some improvements are needed, as expalined in the following session. In any case, the paper is presenting TEM analyses for just one annealing condition, even if for 4 different times.

Response: I am very sorry about it. Owe to the limited project funding, we just finish the TEM analyses of the samples aged at 500 °C and we are planning to do more TEM study, especially on the samples aged at 450°C in the further study.

  1. Punctual comments.
  2. The Abstract is missing one or two rows at the beginning, to introduce the background of this kind of alloys. The authors start in the middle of the action, which is somewhat abrupt for the reader.

Response: We have revised the abstract and introduce the background of these alloys. Please see the revised abstract.

Abstract: Cu-Ni-Si alloys are widely used in electrical and electronic industry owing to excellent electrical conductivity and strength. A suitable addition of Co in the Cu-Ni-Si alloys can improve its strength and deteriorate its electrical conductivity. In this work, Cu-Ni-Co-Si-P-Mg alloys with different Co content are employed to investigate the effects of Co on the properties and microstructure. The results showed that Co addition lead to the formation of (Ni, Co)2Si precipitates. (Ni, Co)2Si precipitate is harder to coarsen than δ-Ni2Si during aging. The larger the Co content in the alloys is, the smaller the precipitates formed is. There exists a threshold content of Co to divide the studied alloys into two groups. One group of theses alloys with < 1 wt. % Co or Co/Ni ratio < 0.56 has the same aging behavior as the Cu-Ni-Si-P-Mg alloy. On the contrary, the time to reach the peak hardness of aging for another group can be obviously postponed and its electrical conductivity decreases slightly with the increase of Co content. It can be attributed to the lower diffusion rate of Co than that of Ni in the Cu matrix. Meanwhile, the Co addition can inhibit the formation of P-enriched Ni-P phase in Co-containing alloys during aging. The as-quenched Cu-1.6Ni-1.2Co-0.65Si-0.1P-0.05Mg alloy can reach 257 HV and 38.7 %IACS after aging at 500 °C for 3h, respectively.

  1. page 2: Please, replace the word «Experimental» with «Materials and methods»

Response: We have replaced “Experimental» with «Materials and methods”.

  1. Page 2: Please, in the description of the alloy fabrication, provide the following details: Which are the details of the raw materials used to fabricate the alloys? size, chemical composition, impurities, etc. Which was the atmosphere in which the alloys were rolled? Why the thickness reduction was different for each chemical composition, and not the same for all the alloys? Which was the starting thickness for all the alloys? How the chemical composition of the alloys was checked, as the author speak just about nominal chemical composition? Which was the preparation surface for hardness (even if it should be addressed to as microhardness) and electrical conductivity?

Response: In the revised manuscript, we have added the details of the raw materials. The hot-rolling and cold rolling was conducted in the air atmosphere. Because our rolling machine is a rough rolling one, it is difficult to control the finishing thickness of the alloy plate so that the thickness reduction was different for these alloys with different chemical composition. We did not analysis the chemical composition of these alloys. All these samples used for hardness and electrical conductivity test were grinded with 2500 grid paper in the end.  

  1. Figure 1 and 2 are missing standard deviations, or any other measure of the dispersion of any data. Sometimes values are very close, so it is not possible to understand if the measure is consistent or not, and if the comparison is misleading or not.

Response: We have add error bar in Fig1 and 2.

  1. page 2: «This indicates the partial substitution of Co for Ni delays significantly the age-hardening response of the alloy.» and also «This suggests the substitution of Co for Ni can not only improve the hardness of the alloy, but also can lower the coarsening rating of the precipitates in these alloys.»
    The author should comment about the fact that it seems that there is some kind of threshold value of Co, or Co and Ni, as the alloys can divided into two groups, each one with a specific behavior: C0 and C1 on one side, and C2, C3 and C4 on the other side. This is specifically clear in Fig. 1a.

Response: We agree with you about this point. Thank you sincerely for your valuable advice. We supplied a paragraph to deal with it. Please see page 4.” In summary, according the aging behavior, the alloys can be divided into two groups. C0 and C1 with Co/Ni weight ratio of 0.27 have the same aging behavior and thus can be classify to one group. C2-C4 alloys with more than 0.6 wt % Co (Co/Ni weight ratio > 0.27) can be classified to another one. In addition, C1 alloy can obtain higher peak hardness than Cu-2.8Ni-0.65Si -0.1P-0.05Mg alloy without Co and almost the same electrical conductivity. Addition of Co (more than 0.6 wt %, or Co/Ni weight ratio > 0.27) can postpone the time of peak hardness during aging. Compared with the studied alloys aged at 450°C, the classification for these alloys aged at 500°C is more obvious. The as-quenched Cu-Ni-Co-Si-P-Mg alloys with Co have the higher hardness than Cu-Ni-Si-P-Mg alloy and a small decrease in hardness after a prolonged aging. It revealed that the alloy containing Co has the excellent thermal stability. When the Co content is larger than 0.6 (Co/Ni weight ratio > 0.27), the effect of Co on improving hardness was impaired and the electrical conductivity decreased with the increase of Co content instead. “

 “

  1. page 2 «It is well known that the conductivity depends on the rate of transport of solute atoms from the matrix to the precipitates.» it is not clear what the authors want to say here; it seems that conductivity is related to solute atoms movement, which is clearly not possible. Please, reorganize this sentence in a better way.

Response: In order not to mislead the readers, we deleted this sentence.

  1. Please, provide indexing of Fig. 3b (inlet), Fig. 3f (inlet), Fig. 4(b) (inlet), Fig. 4(d) (inlets of FFT), Fig. 7 (inlets of FFT), Fig. 8(b). In some diffraction inlets, there are other patterns that need to be better assessed, for example in fig. 3b, 3f, 4a, 4b, 6a, 6f, 7b.

Response: We supplied the indexing of these inset in these figures.

  1. Please, provide a higher count spectrum for Fig.8(d), as there, the count number is very low.

Response: Fig.8(d) has the low count intensity because the particle is close to the rim of hole and very thin so that the intensity of emitted Characteristic X-ray signal is very weak. We replaced it with a new EDS.

  1. Figure 5. Why is P not present? Which is the role of Mg and P in the alloy? Are they not impurities for the considered amounts?

Response: We can found from the EDS of Area#1 (See Figure 5(f)) that P is present in fact. I am very sorry that I can the P mapping image because the TEM operator forgot to take P mapping image due to his operation errors. P in the alloy have three roles: (a) to deoxidize during melting; (b) to improve the mobility of alloy melt; (c) to form the Ni-P precipitates during aging, The residual P in the copper-based solid solution can impair the electrical property of copper alloy. Mg plays two role in the studied alloy: (a) to deoxidize during melting; (b) to improve the mechanical property (for example, resistance to stress relaxation) of alloy due to the Mg-atom-drag effect on dislocation motion as mentioned in the revised manuscript.

  1. Use of English.English needs to be improved, as there are expression that are not acceptable in current use. Here you are a few example; other ones can be found easily in the text.

page 1: «then leads to a higher mechanical property» should be replaced by «then leads to higher mechanical properties»

Response: We have corrected it.

page 1: «did not explain that why the addition of Co» should be replaced by «did not explain why the addition of Co»

Response: We have corrected it.

page 2: «small addition of Co on the propertied and microstructure» should be «properties», not «propertied»

Response: We have corrected it.

page 2: «the addition of Co to the alloy enhances the hardness of the alloy» too many times alloy, here

Response: Sorry. We have deleted the unnecessary “alloy”

page 2: «its electrical conductivity. And the reason for» «And» is not admissible after a full stop. In particular, this kind of mistake is rather frequent in the text. Please, correct it everywhere.

page 13: «Therefore, when the mean sizes of precipitates in the Co-containing aged alloy are smaller than that in the Co-free aged alloy. And the mean size of precipitates decreases with the increase of Co content in the studied alloys.» Please, reformulate the sentences, as they are not understandable in the present form.

Response: Sorry. We have revised this sentence. See the page 14. “Therefore, when the mean sizes of precipitates in the Co-containing aged alloy are smaller than that in the Co-free aged alloy. The higher the Co content in these alloys is, the smaller the mean size of precipitates is.”

Reviewer #2:

  1. Summary.The paper deals with the fabrication of Cu-based alloys for electric and electronic industry. In particular, the paper deals with the analysis of the mechanical, electrical and microstructural features of five alloys, with varying Co and Ni content. The alloys were studied in the as-cast state, after a series of annealing and in the condition of mechanical deformation, obtained by cold rolling. The mechanical properties were studied by microhardness studies, while the electrical properties were assessed by conductivity studies. Transmission electron investigations were carried out to provide an interpretation key for the mechanical and electrical behaviour of the alloys. The subject of the paper is compatible with the topics covered by MDPI «Materials». The work is well organized, even if some major modifications are required before publication. The present document describes the required modifications to be brought to the manuscript to improve it until the final version. Three main subject are covered, that is (1) General comments; (2) Punctual comments, and (3) Use of English.
  2. General comments. The paper is organized in a logic way. The performed characterizations are consistent with the given explanation. Data are presented in a well-organized way, but some improvements are needed, as expalined in the following session. In any case, the paper is presenting TEM analyses for just one annealing condition, even if for 4 different times.

Response: I am very sorry about it. Owe to the limited project funding, we just finish the TEM analyses of the samples aged at 500 °C and we are planning to do more TEM study, especially on the samples aged at 450°C in the further study.

  1. Punctual comments.
  2. The Abstract is missing one or two rows at the beginning, to introduce the background of this kind of alloys. The authors start in the middle of the action, which is somewhat abrupt for the reader.

Response: We have revised the abstract and introduce the background of these alloys. Please see the revised abstract.

Abstract: Cu-Ni-Si alloys are widely used in electrical and electronic industry owing to excellent electrical conductivity and strength. A suitable addition of Co in the Cu-Ni-Si alloys can improve its strength and deteriorate its electrical conductivity. In this work, Cu-Ni-Co-Si-P-Mg alloys with different Co content are employed to investigate the effects of Co on the properties and microstructure. The results showed that Co addition lead to the formation of (Ni, Co)2Si precipitates. (Ni, Co)2Si precipitate is harder to coarsen than δ-Ni2Si during aging. The larger the Co content in the alloys is, the smaller the precipitates formed is. There exists a threshold content of Co to divide the studied alloys into two groups. One group of theses alloys with < 1 wt. % Co or Co/Ni ratio < 0.56 has the same aging behavior as the Cu-Ni-Si-P-Mg alloy. On the contrary, the time to reach the peak hardness of aging for another group can be obviously postponed and its electrical conductivity decreases slightly with the increase of Co content. It can be attributed to the lower diffusion rate of Co than that of Ni in the Cu matrix. Meanwhile, the Co addition can inhibit the formation of P-enriched Ni-P phase in Co-containing alloys during aging. The as-quenched Cu-1.6Ni-1.2Co-0.65Si-0.1P-0.05Mg alloy can reach 257 HV and 38.7 %IACS after aging at 500 °C for 3h, respectively.

  1. page 2: Please, replace the word «Experimental» with «Materials and methods»

Response: We have replaced “Experimental» with «Materials and methods”.

  1. Page 2: Please, in the description of the alloy fabrication, provide the following details: Which are the details of the raw materials used to fabricate the alloys? size, chemical composition, impurities, etc. Which was the atmosphere in which the alloys were rolled? Why the thickness reduction was different for each chemical composition, and not the same for all the alloys? Which was the starting thickness for all the alloys? How the chemical composition of the alloys was checked, as the author speak just about nominal chemical composition? Which was the preparation surface for hardness (even if it should be addressed to as microhardness) and electrical conductivity?

Response: In the revised manuscript, we have added the details of the raw materials. The hot-rolling and cold rolling was conducted in the air atmosphere. Because our rolling machine is a rough rolling one, it is difficult to control the finishing thickness of the alloy plate so that the thickness reduction was different for these alloys with different chemical composition. We did not analysis the chemical composition of these alloys. All these samples used for hardness and electrical conductivity test were grinded with 2500 grid paper in the end.  

  1. Figure 1 and 2 are missing standard deviations, or any other measure of the dispersion of any data. Sometimes values are very close, so it is not possible to understand if the measure is consistent or not, and if the comparison is misleading or not.

Response: We have add error bar in Fig1 and 2.

  1. page 2: «This indicates the partial substitution of Co for Ni delays significantly the age-hardening response of the alloy.» and also «This suggests the substitution of Co for Ni can not only improve the hardness of the alloy, but also can lower the coarsening rating of the precipitates in these alloys.»
    The author should comment about the fact that it seems that there is some kind of threshold value of Co, or Co and Ni, as the alloys can divided into two groups, each one with a specific behavior: C0 and C1 on one side, and C2, C3 and C4 on the other side. This is specifically clear in Fig. 1a.

Response: We agree with you about this point. Thank you sincerely for your valuable advice. We supplied a paragraph to deal with it. Please see page 4.” In summary, according the aging behavior, the alloys can be divided into two groups. C0 and C1 with Co/Ni weight ratio of 0.27 have the same aging behavior and thus can be classify to one group. C2-C4 alloys with more than 0.6 wt % Co (Co/Ni weight ratio > 0.27) can be classified to another one. In addition, C1 alloy can obtain higher peak hardness than Cu-2.8Ni-0.65Si -0.1P-0.05Mg alloy without Co and almost the same electrical conductivity. Addition of Co (more than 0.6 wt %, or Co/Ni weight ratio > 0.27) can postpone the time of peak hardness during aging. Compared with the studied alloys aged at 450°C, the classification for these alloys aged at 500°C is more obvious. The as-quenched Cu-Ni-Co-Si-P-Mg alloys with Co have the higher hardness than Cu-Ni-Si-P-Mg alloy and a small decrease in hardness after a prolonged aging. It revealed that the alloy containing Co has the excellent thermal stability. When the Co content is larger than 0.6 (Co/Ni weight ratio > 0.27), the effect of Co on improving hardness was impaired and the electrical conductivity decreased with the increase of Co content instead. “

 “

  1. page 2 «It is well known that the conductivity depends on the rate of transport of solute atoms from the matrix to the precipitates.» it is not clear what the authors want to say here; it seems that conductivity is related to solute atoms movement, which is clearly not possible. Please, reorganize this sentence in a better way.

Response: In order not to mislead the readers, we deleted this sentence.

  1. Please, provide indexing of Fig. 3b (inlet), Fig. 3f (inlet), Fig. 4(b) (inlet), Fig. 4(d) (inlets of FFT), Fig. 7 (inlets of FFT), Fig. 8(b). In some diffraction inlets, there are other patterns that need to be better assessed, for example in fig. 3b, 3f, 4a, 4b, 6a, 6f, 7b.

Response: We supplied the indexing of these inset in these figures.

  1. Please, provide a higher count spectrum for Fig.8(d), as there, the count number is very low.

Response: Fig.8(d) has the low count intensity because the particle is close to the rim of hole and very thin so that the intensity of emitted Characteristic X-ray signal is very weak. We replaced it with a new EDS.

  1. Figure 5. Why is P not present? Which is the role of Mg and P in the alloy? Are they not impurities for the considered amounts?

Response: We can found from the EDS of Area#1 (See Figure 5(f)) that P is present in fact. I am very sorry that I can the P mapping image because the TEM operator forgot to take P mapping image due to his operation errors. P in the alloy have three roles: (a) to deoxidize during melting; (b) to improve the mobility of alloy melt; (c) to form the Ni-P precipitates during aging, The residual P in the copper-based solid solution can impair the electrical property of copper alloy. Mg plays two role in the studied alloy: (a) to deoxidize during melting; (b) to improve the mechanical property (for example, resistance to stress relaxation) of alloy due to the Mg-atom-drag effect on dislocation motion as mentioned in the revised manuscript.

  1. Use of English.English needs to be improved, as there are expression that are not acceptable in current use. Here you are a few example; other ones can be found easily in the text.

page 1: «then leads to a higher mechanical property» should be replaced by «then leads to higher mechanical properties»

Response: We have corrected it.

page 1: «did not explain that why the addition of Co» should be replaced by «did not explain why the addition of Co»

Response: We have corrected it.

page 2: «small addition of Co on the propertied and microstructure» should be «properties», not «propertied»

Response: We have corrected it.

page 2: «the addition of Co to the alloy enhances the hardness of the alloy» too many times alloy, here

Response: Sorry. We have deleted the unnecessary “alloy”

page 2: «its electrical conductivity. And the reason for» «And» is not admissible after a full stop. In particular, this kind of mistake is rather frequent in the text. Please, correct it everywhere.

page 13: «Therefore, when the mean sizes of precipitates in the Co-containing aged alloy are smaller than that in the Co-free aged alloy. And the mean size of precipitates decreases with the increase of Co content in the studied alloys.» Please, reformulate the sentences, as they are not understandable in the present form.

Response: Sorry. We have revised this sentence. See the page 14. “Therefore, when the mean sizes of precipitates in the Co-containing aged alloy are smaller than that in the Co-free aged alloy. The higher the Co content in these alloys is, the smaller the mean size of precipitates is.”

Reviewer 3 Report

Overall, the work is very interesting, and as it happens with a set of TEM micrographs, it looks good.

Introduction should be in-depth and supplemented with already known information on the role of P and Mg additives. The influence of P is given in the works [16, 21-23], so one should refer to them and provide some detail, for example what kind of separations it creates. Nothing is known about the influence of Mg, and rather none of the cited works concerns an alloy with Mg. Therefore, one should try to justify why it was decided to alloy with both these additives, the more so as no magnesium was found in the precipitates.

When describing the method of sample preparation by rolling and heat treatment, it would be better to use a graphical diagram. It would also be good to mention why the second rolling and aging was performed? Is it an original proposition based on own experience for these alloys, or is it a standard procedure for these alloys by other authors in order to increase the dispersion of the precipitates? About the fact that "Cold rolling accelerated significantly the precipitation process." was written only in application 4. If the authors have photomicrographs of alloys made on an optical microscope or SEM, we can propose two examples, with and without Co.

Conclusion 1 and 4 - there is no need to mention the individual hardnesses, the more that the conclusions should not use the C0-C4 determinations, but refer to the Co concentration/rolling and temperature/time "The hardness values ​​of the C0 and C1 alloys increase to the peak of 235HV and 251HV after aging at 500°C for 1h and 1.5h, respectively, while the hardness values ​​of the C2, C3 and C4 alloys increase to the peak values ​​of 235HV , 257HV 247HV after aging for 1.5h, 3h and 3h, respectively. " - is not a conclusion.

Author Response

Reviewer #3:

Overall, the work is very interesting, and as it happens with a set of TEM micrographs, it looks good.

Introduction should be in-depth and supplemented with already known information on the role of P and Mg additives. The influence of P is given in the works [16, 21-23], so one should refer to them and provide some detail, for example what kind of separations it creates. Nothing is known about the influence of Mg, and rather none of the cited works concerns an alloy with Mg. Therefore, one should try to justify why it was decided to alloy with both these additives, the more so as no magnesium was found in the precipitates.

Response: We have supply the information on the roles of P and Mg; also see the response to the comment of reviewer 3#. The role of Mg in this alloy is to improve the mechanical property (for example, resistance to stress relaxation) of alloy due to the Mg-atom-drag effect on dislocation motion as mentioned in the revised manuscript, and to eliminate the oxygen in melt.

 When describing the method of sample preparation by rolling and heat treatment, it would be better to use a graphical diagram. It would also be good to mention why the second rolling and aging was performed? Is it an original proposition based on own experience for these alloys, or is it a standard procedure for these alloys by other authors in order to increase the dispersion of the precipitates? About the fact that "Cold rolling accelerated significantly the precipitation process." was written only in application 4. If the authors have photomicrographs of alloys made on an optical microscope or SEM, we can propose two examples, with and without Co.

Response: We have supply a table to describe the two process route in table 1. The second rolling (cold rolling) and aging can increase the strength and electrical conductivity of the studied alloys and is standard procedure for these alloys. So it is not specially emphesised. The optical microscope or SEM can be used to observe the difference in microstructure between the two alloys. So we did not do these works.

Conclusion 1 and 4 - there is no need to mention the individual hardnesses, the more that the conclusions should not use the C0-C4 determinations, but refer to the Co concentration/rolling and temperature/time "The hardness values ​​of the C0 and C1 alloys increase to the peak of 235HV and 251HV after aging at 500°C for 1h and 1.5h, respectively, while the hardness values ​​of the C2, C3 and C4 alloys increase to the peak values ​​of 235HV , 257HV 247HV after aging for 1.5h, 3h and 3h, respectively. " - is not a conclusion.

Response: Thanks for your good suggestions. We have corrected the doing according your suggestions.

Conclusions

(1)According the aging behavior, the studied alloys can be divided into two groups. Cu-2.8Ni-0.65Si-0.1P-0.05Mg and Cu-2.2Ni-0.6Co-0.65Si-0.1P-0.05Mg with Co/Ni weight ratio of 0.27 have the same aging behavior, but Cu-2.2Ni-0.6Co-0.65Si-0.1P-0.05Mg can obtain higher peak hardness than the former and almost the same electrical conductivity. For another group of Cu-(2.8-x)Ni-xCo-0.65Si-0.1P-0.05Mg (x≥1 or Co/Ni weight ratio ≥ 0.56) alloys, the aging time to reach the peak hardness can be postponed during aging. However, its electrical conductivity decreases slightly with the increase of Co content.

(2)The addition of Co can reduce the size of precipitates. The larger the amount of Co in the alloys is, the smaller the precipitates formed is. Meanwhile,the addition of Co in the Cu-Ni-Si-P-Mg alloy can also inhibit the formation of P-enriched phase in the Co-containing alloys during aging. The P-enriched phase has a definite orientation relationship with the matrix, which habit plane is parallel to the {001} planes of Cu matrix. The P-enriched precipitate phase is obviously easier to coarsen than the δ-Ni2Si and δ-(Ni, Co)2Si, which results in a greater decrement in hardness for Co-free alloy during over-aging.

(3)The optimal addition amounts of Ni and Co are 1.6 wt. % Ni and 1.2 % Co. The hardness and electrical conductivity of the as-quenched Cu-1.6Ni-1.2Co-0.65Si-0.1P-0.05Mg alloy increase to 257 HV and 38.7 %IACS after aging at 500 °C for 3h, respectively.

Round 2

Reviewer 1 Report

Paper can be accepted in present form

Reviewer 2 Report

No other comments. The paper is finally suitable for publication after a moderate English revision. Thank you.